# Incidence of Burnout Syndrome among Anesthesiologists and Intensivists in France: The REPAR Study

**DOI:** 10.3390/ijerph20031771

**Published:** 2023-01-18

**Authors:** Barnabé Berger, Pierre-Julien Cungi, Ségolène Arzalier, Thomas Lieutaud, Lionel Velly, Pierre Simeone, Nicolas Bruder

**Affiliations:** 1Department of Anaesthesiology and Critical Care Medicine, University Hospital Timone, AP-HM, Aix Marseille University, F-13005 Marseille, France; 2Fédération Anesthésie Réanimation, Hôpital d’Instruction des Armées Sainte Anne, F-83000 Toulon, France; 3Département d’Anesthésie–Réanimation, University Hospital of Caen, Avenue Côte-de-Nacre, F-14000 Caen, France; 4Comité Vie Professionnelle Santé au Travail (CVP-ST), Société Française d’Anesthésie-Réanimation (SFAR), 74, Rue Raynouard, F-75016 Paris, France; 5UMRESTTE, UMR-T9405, Université Gustave Eiffel, Université Claude Bernard de Lyon, CEDEX, F-69675 Bron, France; 6CNRS, Institut de Neurosciences de la Timone, Aix Marseille University, UMR7289, F-13005 Marseille, France

**Keywords:** burnout syndrome, mental health, healthcare practitioners, depression, quality of care, risk management

## Abstract

Background: Burnout syndrome (BOS) impacts health workers and has become a real public health issue. The primary objective of this observational study was to re-evaluate the incidence of BOS among anesthesiologists and intensivists (AI) in France, ten years after the SESMAT study, a French Physician Health Survey carried out among burnout salaried AI. The secondary objective was to investigate risks factors. Methods: The REPAR survey is an observational study carried in France among AI, residents, and seniors, whatever their main mode of practice, in the framework of a self-questionnaire distributed on the Internet from 11 April 2018 to 1 July 2018. BOS was assessed using the Copenhagen Burnout Inventory (CBI). A score above 50% on two of the dimensions (personal burnout and work-related burnout) indicated BOS, as a main criterion. In order to investigate risks factors, questions were about sociodemographic characteristics, professional and extraprofessional environments, personality and mood using the Major Depression Inventory (MDI). A significance threshold of *p* < 0.05 was retained. Results: Among 1519 questionnaires received, 1500 completed questionnaires were analyzed. There were 775 men (52%) and 721 women (48%), aged 23 to 74 years. Among those, 24% suffered from BOS, 9% showed depressive symptoms (MDI > 25) and 35% were considering changing jobs or stopping their studies. There was no significant difference with the SESMAT study for the general BOS score. After multivariate analysis, 12 variables were significantly associated with the main criterion. Conclusions: Ten years after the SESMAT study, the incidence of BOS in AI has not decreased in the largest cohort of AI studied to date in France.

## 1. Introduction

It is now widely accepted that burnout syndrome (BOS) is a serious state that impacts workers at an individual and collective level, as the World Health Organization included it in the 11th Revision of the International Classification of Diseases (ICD-11) as an occupational phenomenon [1]. The heterogeneity of measurement scales defining BOS in the literature has led to non-relevant variations in the estimation of its incidence, because authors use different scales or a different cut-off among the same scale. At the individual level, it is associated with increased risk of addiction, illegal drug use, anxiety, depression, and suicide [2,3,4,5,6]. It is also related to multiple somatic disorders, from sleep disorders to increased cardiovascular risk [7,8]. At a collective level, BOS is a real public health issue associated with absenteeism and job cessation [9,10]. It is well known that health care workers with BOS are at risk of delivering poorer quality of care, and consequently, decreasing quality and safety of patient management. [5,11,12,13,14].

However, few studies have looked at BOS specifically in anesthesiologists and intensivists (AI) in France [4,15,16]. In 2008, in France, the «SESMAT» Physician Health Survey studied the frequency of BOS among senior salaried physicians and pharmacists in hospital. In this study, the Copenhagen Burnout Inventory (CBI) revealed a score of BOS in 38.4% in AI. Over the last ten years, demographic data showed an older and more feminized medical population, while the overall active workforce was stable [17].

Over the last fourteen years, the medical and social environment has changed. BOS and its consequences have been extensively discussed in scientific meetings, workshops and recognized as an important aspect of medical work by French authorities. Several institutions have organized pathways to provide psychological support to physicians who need it. Preventive measures appeared with educational interventions targeting physicians’ self-confidence or communication skills, discussion groups, and duty-hour restrictions, as an excess of working hours above 48 h per week as recommended by European guidelines has been more frequently scrutinized, especially for residents [18].

Whether this awareness on the impact of BOS and preventive measures have been effective or not is unknown, and if not, identifying risks factors may lead to a better understanding of BOS among AI and therefore potentially lead to better care for practitioners. Ten years after the SESMAT study, we were unable to follow the same group of AI, so we designed the REPAR study to reassess the incidence and risk factors for BOS and the changes over the last 10 years.

## 2. Materials and Methods

### 2.1. Study Design

The REPAR survey is an observational study carried out in France in the framework of a self-questionnaire. This questionnaire was distributed on the Internet with the support of the French Society of Anesthesia and Resuscitation (SFAR), the French College of Anesthesia and Resuscitation (CFAR), regional university representatives, and unions, using their Internet sites, social networks, and mailing lists of their members. The target population of this study was seniors and residents AI practicing in France whatever their location or their main mode of practice.

The questionnaire was developed by senior and resident anesthesiologists and a psychiatrist. To determine questions, we relied on a literature review to look for risk factors and prevention axes already identified. This review was carried out using searches on the PubMed, Embase, Google Scholar and Medline databases and was limited to French and English. This review analyzed studies and works published before December 2017. The questionnaire consisted of 88 questions divided into six categories:-BOS assessment by CBI;-Mood questionnaire with the Major Depression Inventory (MDI) [19];-Sociodemographic characteristics;-Professional environment;-Social environment, behavior, and personality trait;-Opinion survey with the opportunity to leave a free comment.

The numerical scales were standardized from 1 to 10. To reduce the incidence of missing data, we chose to make it mandatory to answer questions related to our main objective. We used the CBI translation used in the SESMAT study, and the French translation of the MDI [16,20]. Other questions were designed by the authors, based upon existing scientific literature (Appendix A).

### 2.2. External Validity

To assess our sample representativity in relation to the population of AI in France, we planned to compare their main socio-demographic characteristics and their mode of practice with the data of the Conseil National de l’Ordre des Médecins. We also planned to compare our results with those of the SESMAT study using the same criteria as those of this study, i.e., a presence of BOS for a CBI score higher than 50%.

### 2.3. Study Endpoints

BOS was studied using the Copenhagen Burnout Inventory (CBI), a self-administered questionnaire with 19 questions in three sections [21]. The first part evaluates “personal burnout”: this is the degree of physical and psychological fatigue and exhaustion experienced by the person. The second part determines the degree of physical and psychological fatigue and exhaustion that is perceived by the person as related to his/her work: defined as “work-related” burnout”. The third part specifies “patient-related burnout”. For each question, five response categories are possible, representing a scale of frequency or intensity, ranging from “never or almost never or very little” to “always or very much”. This scale is reduced to a percentage, without any notion of threshold, with 100% expressing a maximum level of BOS. The CBI is free of charge and validated by the “Haute Autorité en Santé”, the French regulatory agency for healthcare.

In conformity with the usage guidance of Kristensen et al. [21], we analyzed each of the three parts of the CBI separately in percentage format and we chose a cut-off limit for our main outcome measure that was meaningful and easily reproductible: we considered that people had BOS if he or she scored above 50% on two of the dimensions: personal burnout and work-related burnout. For the record, and as explicitly mentioned in its original description [21], CBI scales do not measure stable traits of the individuals but degrees of burnout that may change overtime and the use of a dual criterion makes it possible to eliminate non-work-related burnout states. For statistical analyses using a continuous numeric scale, we used a percentage value.

CBI was used in 2008 by Doppia et al. in the SESMAT study [16] to assess the incidence of BOS among employed physicians and pharmacists in health care facilities via an anonymous self-report questionnaire distributed online. In this study, the CBI results of “personal burnout” and “patient-related” categories are placed into two classes: low burnout if <3 (i.e., 50% in the original CBI) or high burnout if ≥3. Therefore, we planned to compare our CBI results with the SESMAT study results limiting our sample to the same population with people working at a public hospital.

### 2.4. Statistical Analysis

Data were tested for normality (Kolmogorov–Smirnov test) and are presented as mean (standard deviation) (SD) or median (interquartile range “Q1; Q3”) for continuous variables when appropriate. Categorical variables are presented as n (%). Consistency with CBI categories was assessed by Cronbach’s alpha index. Univariate analysis was initially performed to search for risk factors (Student’s t-test or Wilcoxon Mann–Whitney test for quantitative variables and Chi² or Fisher test for qualitative variables). Multivariate analysis was then carried out by logistic regression according to the stepwise descending likelihood ratio model, including the variables linked in univariate to BOS criterion with a *p* < 0.20 threshold. A significance threshold of *p* < 0.05 was retained. The validity of the model was tested by a Hosmer–Lemeshow test and the ROC of the area under the curve. All statistics were performed with IBM SPSS^©^ Statistics 25.0 software.

### 2.5. Standard Protocol Approvals, Registrations, and Patients Consents

The questionnaire was anonymous. An e-mail address was requested to avoid duplicate responses. The consent of each participant was obtained. Data collection followed the recommendations of the CNIL (Commision national de l’informatique et des libertés) and participants were given the legal information concerning the collection of personal data (French law “informatique et libertés” and RGPD “Règle Générale pour la Protection des Données Personnelles”). In accordance with French law, there was no need for the approval of an ethic committee for this non-interventional descriptive study and people were informed of their initial inclusion in the database.

## 3. Results

### 3.1. Participants of the REPAR Study and Their Characteristics

We collected 1519 questionnaires from 11 April 2018 to 1 July 2018. We excluded a total of 19 questionnaires that were duplicates or responses from non AI. Thus, we analyzed 1500 completed questionnaires, representing nearly 11% of the targeted residents and 12% of the targeted seniors. Due to the online diffuse of the questionnaire using both mailing lists and social networks, with multiple receptions possible, the precise response rate could not be determined. The proportion of residents and seniors was not significantly different from that of the target population (16% vs. 18%; *p* = 0.10). A total of 13% (n = 188) responded to the questionnaire the day after an on-call. The distribution of professional profiles is presented in the Appendix A. Our sample was composed of 775 men (52%) and 721 women (48%), aged 23 to 74 years, including 254 residents (17%). The mean age was 43 ± 13 years.

### 3.2. BOS Outcomes

Three hundred and sixty-six AI (24.4%) had personal and work-related burnout scores above 50%. Twenty physicians (1%) had personal and work-related burnout scores above 75%. The mean personal burnout score was 43 ± 24%. One third of respondents (35%, n = 521) had a score above 50% and 86 (5.7%) had a score above 75%. The mean work-related burnout score was 40 ± 26%; 441 (25%) had a score above 50% and 65 (4%) had a score above 75%. The mean patient-related burnout score was 23 ± 24%; 141 (9%) had a score above 50% and 15 (1%) had a score above 75%. Detailed CBI scores for our sample are presented in Figure 1.

### 3.3. External Validity

The theoretical target population was 10,127 AI seniors and 2242 residents, following the number of resident positions offered in the last five years according to the figures of the Conseil National de l’Ordre des Médecins (CNOM) [17]. The percentage of senior physicians in private practice in our study population was 29% (n = 356) which was not statistically different from the general population of AI according to the CNOM (34%, (n = 3461; *p* = 0.41)). Comparison of the demographic characteristics of our sample to the CNOM data is presented in Appendix A. In our study, there was no significant difference with the SESMAT study for the general BOS score in 2008 (with residents and private practice AI excluded) (Table 1). There was a significant decrease in the percentage of patient-related burnout (*p* < 0.01).

### 3.4. Risk Factors Related to BOS

The comparison of different items according to the main criterion with univariate analysis is presented in Table 2. After multivariate analysis, 12 variables were significant associated with the criterion “average personal and work-related burnout scores strictly greater than 50%” (Table 3).

### 3.5. Other Results

Five hundred and ninety-eight (40%) of the participants had difficulty controlling their weight and the mean BMI was 24 ± 4. In addition, 136 physicians (9%) had an MDI score above 25. The correlation between CBI score and MDI score was large (R^2^ = 0.52; *p* < 0.001) (Figure 2). Of the 198 divorced AI (13%), 100 (50.5%) attributed their divorce, at least in part, to their work. Concerning personal conflicts (Appendix A), 157 AI had conflicts “several times a week” or “every day or almost every day” (11%) and 343 several times a month (23%). Other results are available in Appendix A.

### 3.6. Subgroups Analysis

The same risk factors for BOS were found in men and women except for parenthood in women (*p* < 0.01) (Appendix A). Personal and work-related burnout scores were significantly lower in private practice AI (*p* < 0.05) (Appendix A). BOS was found in 20% (71) of respondents in private practice versus 26% (232) in public practice (*p* < 0.05).

## 4. Discussion

This 2018 observational survey revealed BOS in 24% of AI practicing in France. This study is comparable to the SESMAT cohort in 2008 [16]. BOS was defined in our study by mean scores of personal and work-related burnout at CBI strictly above 50%. In our study, 1.3% of the population had both scores above 75%, revealing a severe form of BOS and therefore at high risk of serious personal and professional deleterious consequences. The patient-related burnout score was almost twice as low as the other two components of the questionnaire, with an average score of 23%. In other national works, Chiron et al. [15] identified, in 2010, high scores of emotional exhaustion and depersonalization in favor of BOS in 31% and 34% AI, respectively. In addition, Mion et al. [3] concluded in 2013 that BOS was found in 62% of the 1091 AI using the Maslach Burnout Inventory (MBI). These differences are not easily interpretable because of the heterogeneity of the methodology used. Regarding our comparison to the SESMAT results, confounding factors may exist, which could not be adjusted because it was impossible to strictly survey the same participants ten years after. On the other hand, the targeted group and the scale used were the same to optimize external validity. However, the high prevalence of BOS reported in our study shows that there is still an important issue of BOS prevention on both a personal and societal level with better practices to take care of the practitioners. Our study, with a mixed evaluation between validated scales (CBI and MDI) and more open questions assessing phenomena reported in the literature, allows to better appreciate the epidemiological complexity of this syndrome.

We decided to use CBI rather than MBI [22] for the measurement of BOS for several reasons. First, MBI use is subject to copyright. Second, this divided score leads to problems in defining the primary criterion “burnout” across studies, with each study redefining a different criterion, as observed in other works, with a high concern for reproducibility [3,15,23]. Several theoretical criticisms were formulated against MBI in its construction and are at the origin of the work that led to the CBI validated more recently in 2005 [21]. CBI has been, translated into many languages, validated and recommended by the “Guide d’aide à la prevention” published by the INRS (Institut National de Recherche et de Sécurité). Above all, it was used in the SESMAT study, ten years ago, so we could compare our results.

In our study, working time was not related to BOS in multivariate analysis. The average working time did not increase in ten years: In Mion’s study, the average weekly working time was 53 ± 14 h compared to 56 ± 10 h in our survey with identical averages of four night shifts per month [3]. The literature clearly links the number of hours worked to BOS and even to medical errors [24]. However, most of these studies report very high work duration, sometimes more than 80 h per week. In addition, it seems that the management of working time and organization at work are more closely linked to BOS than the number of working hours. In our study, overall dissatisfaction with one’s own work situation, material working conditions (personal space, transportation) and organizational conditions (lack of meal breaks) were significantly related to the occurrence of BOS. The frequency of conflicts, especially with surgeons, were very significantly related to BOS. Intention to leave the profession among surveyed AI was 61% for those with a high BOS score. The average patient-related burnout score has decreased significantly over the past 10 years, while the incidence of BOS has remained stable. The doctor–patient relationship does not appear to be a predominant mechanism in the occurrence of BOS among AI in France. Among the AI who responded to the REPAR survey, we observed that certain social or psychological conditions are significantly linked to the presence of a BOS.

There was a link between female gender and occurrence of BOS, as in the studies by Doppia et al. or Chiron et al. [15,16]. This disparity in most psychological disorders is well described in the literature, without being able to provide an answer as to its origin. According to Papanikola et al. [25], it is probably a mix of genetic and psychosocial factors [23]. Subgroup analysis identified parenthood as a risk factor for the development of BOS in women (*p* < 0.05) but not in men (*p* > 0.2). However, our study was not designed to better understand the relationships between gender and the occurrence of BOS, and the results remain complex to assess regarding this topic.

Our study did not find a significant association between age and BOS. These results are contrary to those of Chiron et al. and Doppia et al. in France [15,16]. This may be a consequence of improved working conditions for residents and junior doctor. Divorced status was significantly related to the presence of BOS and more than half of the divorced AI attributed their divorce at least in part to their work. In the SESMAT study, a high work–family conflict score was a risk factor for BOS [16]. 

The use of a psychotropic treatment was significantly related to BOS. These treatments may be a consequence of BOS or a sign of health psychological impairment [7]. The statistical analysis revealed a strong relationship between the CBI and MDI scores. This finding agrees with the conclusion of a recent meta-analysis confirming a link between BOS and depression by analyzing 67 studies and more than 80,000 participants [26].

In our study, there was a significant relationship between the time spent on leisure activities and the presence of BOS: AI without BOS spent on average 2 h more per week on leisure activities and almost one hour more on sports. It is a coping strategy well described by Zwack and Schweitzer [27] that reflects the ability of physicians to take care of themselves, particularly through a lifestyle dedicating time to extraprofessional activities. We note a significant link between the desire or the taking of a sabbatical period and BOS: this may reflect the need for physical and especially psychological rest [28]. It could also reflect difficulties in readjusting to stress after a period of rest. Another strategy was an ability to focus on positive aspects, to conduct self-reflection to protect themselves. This subjectivity could explain why the AI with BOS also had the feeling of experiencing more difficulty controlling their weight for similar average BMIs between the two groups. The combination of those factors, which are obviously interrelated, may presume that a lifestyle, with a good work-life balance and socially healthy behaviors, at work and away from work, may prevent BOS. The effectiveness of emotion regulation strategies, including meditation, mindfulness, or self-compassion, in the prevention of BOS has been the subject of several studies [7,27,28,29,30,31]. In a meta-analysis including 15 randomized trials and 37 cohort studies, West et al. concluded that these individual measures, like organizational measures, could significantly decrease the incidence of BOS among physicians [32]. A combination of these measures is likely needed to decrease the incidence of BOS among AI in France (Appendix A).

This study has potential limitations. First, the response rate could not be precisely determined, even if the mailing list and social media covered more than half of the targeted population. In order to assess nonresponse bias, we compared our study population to national data about AI in France: This sample was representative of the target population in terms of the type of activity and proportion of seniors and residents. However, there was an over-representation of women and younger age groups. This can be explained by the method of dissemination of our questionnaire via the Internet and social networks, which allowed for many responses (the largest to date in France in this population) while limiting missing data, as some responses were mandatory. Then, we noticed an over-representation of younger age groups but we do not think it might have biased our conclusions as they are not related to burnout anymore in our study. Finally, some questions were not addressed, such as the presence and number of on-call duty, harassment at work, the practice of hypnosis, or the occurrence of medical errors.

## 5. Conclusions

In the REPAR study, we used the CBI with simple and reproducible criterion, on the largest cohort of AI studied to date in France. We found that nearly a quarter of AI suffered from BOS and 1% suffered from a severe form. Ten years after the SESMAT study, despite a focus on BOS syndrome by scientific societies and the health care institution, the incidence of BOS among AI has not decreased.

The COVID crisis has highlighted this phenomenon that existed before the crisis [33]. These are mainly related to the work organization and individual vulnerability. Our work focuses on the pre-COVID period. This is the last assessment realized for AI in France which underlines that they had to face this crisis in a context where a quarter of AI were already in burnout. A new assessment should be carried out to see whether the awareness of this phenomenon has led to an improvement in working conditions and whether the many statutory changes that have occurred since then in the hospital sector have been effective or whether the opposite is true, with a lack of improvement in risk factors. However, this assessment should be made at a distance from this crisis, because of the various studies carried out at the time, which reported abnormally high rates of burnout during a period, which is not a fair representation of the normal activity of health professionals [34] and which could worsen the results in the absence of a cooling-off period [35].

Our study suggests that changes in the organization at work and working environment would be necessary to mitigate the incidence of BOS. Follow-up studies on BOS seem important to assess the effectiveness of these measures and improve physician quality of life, especially after the health COVID crisis, which occurred after our assessment.

## Figures and Tables

**Figure 1 ijerph-20-01771-f001:**
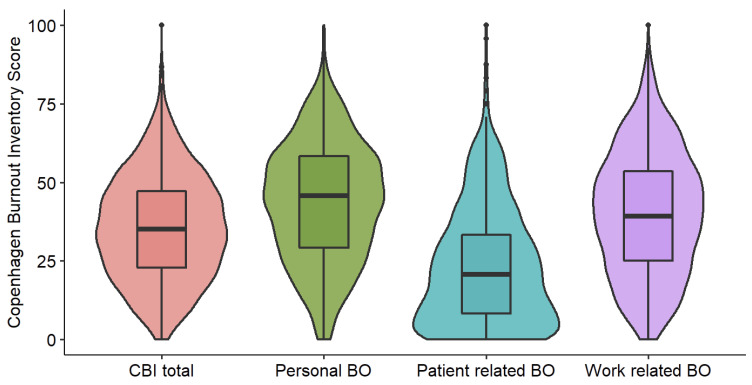
Copenhagen Burnout Inventory (CBI) score with total score and its three components to diagnose burnout syndrome (BO).

**Figure 2 ijerph-20-01771-f002:**
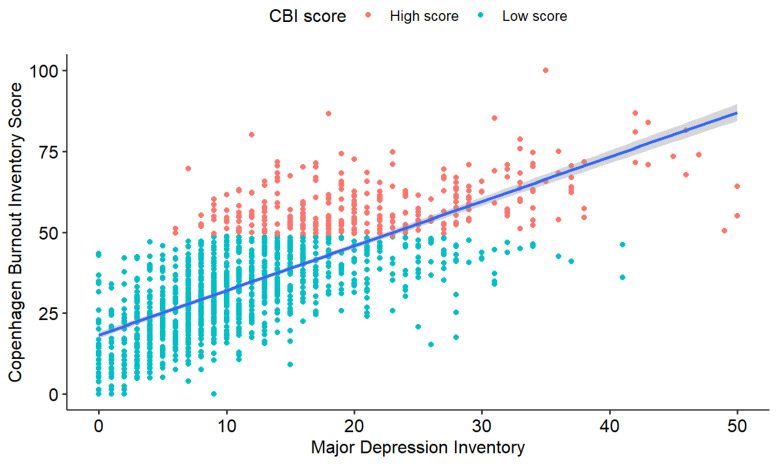
Correlation between Major Depression Inventory (MDI) score and Copenhagen Burnout Inventory (CBI) score among AI in REPAR study, (Linear R² = 0.52; *p* < 0.001).

**Table 1 ijerph-20-01771-t001:** Analysis comparing SESMAT population to REPAR study sample.

	2018	2008	*p*-Value
Personal burnout	890	562	
Low	503 (56.5%)	346 (61.6%)	0.0630
High	387 (43.5%)	216 (38.4%)	
Patient related burnout	890	560	
Low	808 (80.8%)	463 (82.7%)	<0.001
High	82 (9.2%)	97 (17.3%)	

Note: The REPAR sudy sample as been adapted to the SESMAT population, excluding private practice and residents, and this analysis uses the SESMAT low burnout definition: CBI score strictly under 50%.

**Table 2 ijerph-20-01771-t002:** Comparison of different items according to the main criterion regarding burnout syndrome (BOS) with univariate analysis (* *p* < 0.2; *** *p* < 0.05), scales are from 1 to 10.

Item	Total (n = 1500)	No BOS (n = 1134)	BOS (n = 366)	*p*	
Social characteristics					
Mean age	1328	42.8 ± 12.8	42.6 ± 12.3	0.800	
Female gender	1500	44.4%	59.6%	0.000	***
Being in a relationship	1495	80.3%	77.6%	0.2576	
Divorced	1487	11.7%	18.6%	0.0007	***
Medical or paramedical spouse	1234	54.0%	50.0%	0.2254	
Parenting	1494	63.9%	66.7%	0.3394	
Mental Health					
Consulting a psychologist or a psychiatrist on a regular basis	1498	19.3%	26.5%	0.0031	***
Using psychotropic drugs	1499	4.0%	15.0%	0.000	***
MDI > 25	1500	3.2%	27.3%	0.000	***
MDI mean score	1500	9.73 ± 6.6	20.97 ± 9.1	0.000	***
Drugs					
Tobacco (>16 days per month)	1500	11.5%	11.4%	0.8911	
Alcohol (>16 days per month)	1464	22.3%	18.6%	0.1494	*
Cannabis (>16 days per month)	1428	0.6%	1.7%	0.0605	*
Cannabis (even on an occasional basis)	1428	8.4%	7.0%	0.4096	
Benzodiazepines and barbiturates (>16 days per month)	1438	0.8%	2.9%	0.0038	***
Benzodiazepines and barbiturates (even on an occasional basis)	1438	9.2%	17.1%	0.000.	***
Opioids (>16 j/mois)	1431	0.4%	1.5%	0.0408	***
Opioids (even on an occasional basis)	1431	1.7%	3.2%	0.1188	*
Other mind altering drugs and ketamine (even on an occasional basis)	1427	0.9%	0.6%	0.7417	
Cocaïne and amphetamines	1432	1.5%	2.0%	0.4630	
Life outside work					
Lack of sleep during work periods	1496	63.7%	84.7%	0.000.	***
Hours of sleep per night	1498	6.7 h ± 0.9	6.4 h ± 1.0	0.000	***
Spare time scale > 5	1499	36.2%	12.3%	0.000	***
Spare time (without sport)	1454	5.7 h ± 6.5	3.7 h ± 6.3	0.000	***
Regular practice of sport	1496	56.2%	44.0%	0.000	***
Hours dedicated to sport	1412	2.3 h ± 2.2	1.6 ± 1.7	0.000	***
Financial difficulties (often/always)	1498	5.6%	10.9%	0.0004	***
Challenge with weight control	1494	34.8%	56.0%	0.000.	***
BMI	1482	23.9 ± 3.7	24.2 ± 4.0	0.1571	*
Work life					
Day after night-shift answer	1500	12.30%	13.40%	0.5701	
Resident vs. senior doctors	1500	31.1%	27.3%	0.1643	*
Public practice	1246	66.8%	76.6%	0.0228	***
Mean time working per week	1481	56	59	0.000	***
Mean number of nightshifts per month	1490	4.2	4.6	0.003	***
Rest period after night shift	1500	70.5%	72.3%	0.3775	
Semester off (residents)	252	33.2%	45.2%	0.0875	*
Semester off during during residency (seniors)	1227	7.8%	12.3%	0.0157	***
Main practice (seniors): anesthesia/intensive care/both	1239	70.3%/11.5%/18.2%	66.4%/11.6%/21.9%	0.3485	
≥2 stopped work for illness < 14 days in the last 6 months	1496	1.9%	3.3%	0.1373	*
Bad relationships with:					
Paramedical personnel in your department	1493	1.2%	8.0%	0.000	***
Paramedical personnel in other departments	1490	3.1%	9.4%	0.000	***
Doctors in your department	1489	6.1%	19.4%	0.000	***
Doctors in other departments	1488	5.7%	18.3%	0.000	***
Surgeons	1491	15.3%	39.5%	0.000	***
Patients	1492	1.3%	4.1%	0.001	***
Administration	1490	34.0%	53.6%	0.000	***
Many conflicts at work (>1 per months)	1497	27.3%	52.2%	0.000	***
Skills properly recognized scale > 5	1495	71.6%	40.4%	0.000	***
Salary scale > 5	1496	59.0%	34.8%	0.000	***
Working material conditions scale > 5	1497	32.2%	29.0%	0.000	***
Way to work scale > 5	1499	80.5%	64.7%	0.000	***
Personal space at work scale > 5	1498	57.5%	34.0%	0.000	***
Communication equipment at work scale > 5	1499	79.6%	62.3%	0.000	***
Workteam feeling scale > 5	1499	79.8%	54.8%	0.000	***
Freedom of action at work scale > 5	1496	77.3%	47.8%	0.000	***
Work global situation scale > 5	1500	78.1%	31.1%	0.000	***
Team size (median [interquartile])	1465	12 [4–20]	10 [3–20]	0.890	
Seniority within the team (years) (median [interquartile])	1383	5 [2–13]	5 [2–11]	0.568	
Time interval since last vacations (weeks) (median [interquartile])	1478	6 [3–10]	6 [4–12]	0.000	***
Lunch break (night shift excluded): often or always	1400	70.9%	48.8%	0.000	***
Lunch break during night shift: often or always	1496	74.5%	54.1%	0.000	***
Team meeting (>1 per month)	1498	54.9%	40.8%	0.000	***
Time spent on training	1493	51.3%	34.0%	0.000	***
Considering changing job or stopping study (now)	1500	23.8%	62.0%	0.000	***
Having a regular doctor	1495	31.6%	32.6%	0.913	
Occupational health care visit within last 3 years	1497	18.9%	14.8%	0.0840	

**Table 3 ijerph-20-01771-t003:** Multivariate analysis with variables significantly associated with the criterion “average personal and work-related burnout scores strictly greater than 50%”.

	*p*	OR	IC 95%
MDI score > 25	<0.001	5.37	[3.24–8.91]
Work global situation scale ≤ 5	<0.001	4.08	[2.96–5.61]
Using psychotropic drugs	0.001	2.69	[1.53–4.71]
Divorce	<0.001	2.13	[1.40–3.25]
Conflict with surgeons	0.001	1.82	[1.28–2.58]
Year or semester time-off	0.008	1.81	[1.17–2.79]
Frequent workplace conflict	0.001	1.75	[1.27–2.41]
Female gender	0.001	1.69	[1.24–2.31]
No lunch break	0.003	1.63	[1.18–2.24]
Difficult weight control	0.003	1.59	[1.17–2.16]
Difficult work access transport scale	0.047	1.41	[1.00–1.97]
Leisure time (in hours)	0.013	0.96	[0.93–0.99]

## Data Availability

Data available on request due to restrictions eg privacy or ethical.

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
