# Peer review of "Incidence of Burnout Syndrome among Anesthesiologists and Intensivists in France: The REPAR Study"

_ijerph, 2023, doi:10.3390/ijerph20031771_

Round 1
Reviewer 1 Report (Previous Reviewer 3)
In this study, the authors compared the 2018 survey data with the 2008 survey data, which is a more interesting approach. However, 10 years have passed, the participants are not the same population at all, and the results of the comparison involve many confounding factors that cannot be adjusted. This should be described in the limitations
Author Response
We thank the reviewer for the constructive suggestions we have found in the report.
Please find our response in the attached file.

Reviewer 2 Report (New Reviewer)
Thank you for the opportunity to review the manuscript ‘
Incidence of Burnout Syndrome among Anesthesiologists and 2 Intensivists in France: The REPAR Study. ‘
This paper has two aims: The first aim was to investigate the incidence of burn-out symptomatology (BOS) among anesthesiologists and intensivists (AI) in France. The second aim was to take a closer look at risk factors for BOS. This was done by means of an observational study. The abstract generally provides a good overview of the study, but has some areas for improvement:
Title and abstract
The title of the study was chosen concisely and appropriately. The reader gets an idea of the content directly.
The target group of the respondents is not clear. In the results section, it is mentioned that the participants are considering quitting their job or their studies. The authors should briefly mention whether students were (also) surveyed. In addition, the location of the survey should be mentioned for better clarification.
Also, the sentence from line 23 to 25 seems incomplete (maybe "..were assessed" is missing?). Please complete this sentence. In addition, this sentence should be better connected to the overall construct with a preceding explanatory sentence.
Line 27: “9% were depressed”: As a complete diagnosis is not carried out, a different description should be chosen, such as “showed depressive symptoms”
The description of the methodology carried out should be given briefly in the methods section.
Line 29: For a better overview, the main criterion should be mentioned here. The authors could also give examples of some of the 12 factors identified here.
In line 30, the SESMAT study is mentioned; if this is not known to the general readership, it is difficult to establish a connection. Therefore, either the statement of the conclusion should be revised again or the subject of SESMAT should be explained with a side sentence.
Introduction
After the first sentence there should be a citation supporting this statement.
What do the authors mean by “artificial variations” in line 38? The heterogeneity of the measuring instruments and the resulting heterogeneity of the incidences should be addressed more precisely here.
The authors state that the SESMAT study has triggered some crucial changes. It would be helpful for the reader to be provided with specific sources, i.e. which institutions were involved, whether universities were also involved, etc. The changes mentioned should also be accompanied by literature.
At the end of the introduction, the authors should again clearly state which research gap the present work is intended to fill and why the risk factors mentioned in the abstract are of interest.
Materials and Methods
The exact data of the survey listed in the abstract can also be moved to the "study design" section for better replicability.
Moreover, the authors explain that the questionnaires were widely distributed via several websites. It would be interesting to know exactly which advertising sites were used, since a very specific clientele (AI) was to be addressed.
Moreover, was the approval of an ethics committee been obtained? And if not, why has this not been done? Please specify here.
With regard to the questionnaires, the authors state that the questionnaires were developed by the team itself.
With regard to the questionnaires, the authors state that the questionnaires were developed by the team itself.
The authors mention different categories into which the questionnaires were divided. However, CBI and MDI are already existing validated instruments. This section should therefore be revised again to make clear where the authors have created questionnaires themselves and where they have used existing materials.
Results and discussion
Line 144: Which population of AI is being referred to here? Please specify or refer to section 3.3. Also, the numbers in the results section should not be written out.Figure 1: The heading of figure 1 should still be adjusted. Instead of "name", a heading as precise as possible (e.g. total and sub-scales of burnout) should be chosen.
The explanation of table 1 should be moved to a sub-heading according to the APA-guidelines.
Table 2 reports some results whose operationalisation is not comprehensible. For example, which instrument was used to measure drug use? Was it classified under one of the six factors? This should therefore be explained in the methods section.
It remains unclear how the the selection of the reported results under 3.5. was done. Why is the BMI of particular interest for the authors? If this outcome is to continue to be reported, the relationship between weight or BMI and burnout should be explained in the introduction.
Discussion
The discussion should address the implications of the prevalences found at the personal and societal level. The authors mention the heterogeneity of the results found, but the advantages of recording using different instruments could also be mentioned.
The paragraph on working time (from line 244) should be moved to the Limitations, as this no longer focuses on the results of the study in a closer sense. Also, the paragraph on COVID fits more coherently into the conclusion in terms of content and further research need.
The authors also refer to the relationship between gender and mental illness (line261).
The field of discussion hereby opened is too large, therefore, possible correlations and vulnerability and resilience factors with regard to gender in burnout should be dealt with in a more differentiated way.
Line 289: the authors should support the connection between a healthy lifestyle, weight control and burn out with literature.
In addition, the entire manuscript should be checked for spelling and grammar, as well as for APA-conformity.
Author Response
We thank the reviewer for the constructive suggestions we have found in the report.
Please find our response in the attached file.

Reviewer 3 Report (New Reviewer)
Review Why used the Copenhagen Burnout Inventory (CBI). Introduction 4th line - associated with increased risk of suicide, addiction and illegal 39 drug use [1-5]. Describe other mental health issues too - Extraprofessional environment and personality - meaning of that ?? we planned to compare our CBI 118 results with the SESMAT study results - describe the SESMAT studyexplain why factors like life outsider work , consulting psychiatrist , use of cannabis alcohol - are considered and important
why one would face financial difficulties , social structure of a specialist there in the country controlling their weight - explain further
Author Response
We thank the reviewer for the constructive suggestions we have found in the report.
Please find our response in the attached file.

Round 2
Reviewer 2 Report (New Reviewer)
Please see the attached file.

Author Response
Please see the attached file.

Reviewer 3 Report (New Reviewer)
good work
please do similar studies on other professionals too
Author Response
Thank you very much for your advices.
This manuscript is a resubmission of an earlier submission. The following is a list of the peer review reports and author responses from that submission.
Round 1
Reviewer 1 Report
In the introduction, you could in the last paragraph better describe the preventive measures 54 (line 54) to which you refer.
Author Response
Thank you very much for your suggestions regarding our manuscript. Please see the attachment.

Reviewer 2 Report
Burnout is a serious issue across health care providers, especially after the COVID-19 pandemic. I have no issues with the background, methods, results, discussion apart from moderate corrections needed in the use of english. I would recommend that a statistician review all the data analysis and conclusions.
My major hesitancy about this paper is that in my view it is outdated as there has been a flood of literature post pandemic about burnout in healthcare. I think the authors would do well to repeat the online questionnaire now and write about their results pre and post pandemic.
Similarly, the references are stale for a paper due to be published in the fall-winter of 2022.
Author Response

(The authors gave the same response as above.)

Reviewer 3 Report
This study surveyed the prevalence of burnout syndrome among anesthesiologists and critical care physicians in France and revealed interesting details about risk factors.
This manuscript can be improved from the following aspects:
1. Abstract: When AI is displayed in text for the first time, it should be described by its full name.
2. Introduction: The background of this survey need to be explained in detail. After ten years of SESMAT study, we were unable to survey the same group of AI to assess changes in job burnout, so this purpose was not feasible. You could say that the medical and social environment has changed in a decade and we should reassess the situation and risk factors.
3. Methods: Detailed information of scale should be provided, full name, cutoff, sensitivity and specificity in French et al. 2.4 Study end points should be outcome measures.
4. Results: How many questionnaires were sent? Response rate? How is nonresponse bias assessed?
5. Discussion: The mechanisms underlying the results should be fully discussed, for instance obesity being associated with inflammation, anxiety/depression, may increase burnout. Bmi, time of sport and the difficulty of controlling BMI may reflect a lifestyle.
Author Response

(The authors gave the same response as above.)

Round 2
Reviewer 2 Report
I think this manuscript could be resubmitted with results from a post COVID repeat survey with a review of recent literature on the subject